

# How many ontogenetic points are needed to accurately describe the ontogeny of a cephalopod conch? A case study of the modern nautilid *Nautilus pompilius*

Amane Tajika[1] and Christian Klug[2]

[1] Division of Paleontology (Invertebrates), American Museum of Natural History, New York, United States of America

[2] Paläontologisches Institut und Museum, Universität Zürich, Zurich, Switzerland

## ABSTRACT

Recent advancements in tomographic techniques allow for detailed morphological analysis of various organisms, which has proved difficult in the past. However, the time and cost required for the post-processing of highly resolved tomographic data are considerable. Cephalopods are an ideal group to study ontogeny using tomography as the entire life history is preserved within a conch. Although an increasing number of studies apply tomography to cephalopod conchs, the number of conch measurements needed to adequately characterize ontogeny remains unknown. Therefore, the effect of different ontogenetic sampling densities on the accuracy of the resultant growth trajectories needs to be investigated. Here, we reconstruct ontogenetic trajectories of a single conch of *Nautilus pompilius* using different numbers of ontogenetic points to assess the resulting accuracies. To this end, conch parameters were measured every 10°, 30°, 45°, 90°, and 180°. Results reveal that the overall patterns of reconstructed growth trajectories are nearly identical. Relatively large errors appear to occur where growth changes occur, such as the points of hatching and the onset of morphogenetic countdown before the attainment of maturity. In addition, a previously undocumented growth change before hatching was detected when measurements were taken every 10°, 30°, and 45°, though this growth change was obscured when fewer measurements were used (90° and 180°). The lower number of measurements also masks the subtle fluctuating patterns of conch parameters in middle ontogeny. We conclude that the measurements of a conch every 30° and 45° permit a reasonably precise description of conch ontogeny in nautilids. Since ammonoids were likely more responsive to external stimuli than to nautilids, a much denser sampling may be required for ammonoids.

# INTRODUCTION

The advancements of various tomographic techniques have enabled high-resolution reconstructions of diverse objects. In palaeontology, the number of studies using computed tomography (CT) has increased dramatically over past decades, having been referred to as the "CT revolution" (*Sutton, Rahman & Garwood, 2014*). Acquisition of highly resolved

Corresponding author
Amane Tajika, atajika@amnh.org

raw data using tomographic techniques has become relatively easy, whereas post-processing of the acquired data is still time-consuming and often expensive (especially hardware and software).

Cephalopods are an ideal model group to test hypotheses in important fields of palaeontology. One of the advantages of cephalopod conchs is that the entire ontogeny is preserved within a conch. An increasing number of studies apply both destructive and non-destructive tomographic methods to access various aspects of morphology and ontogeny of cephalopod conchs and to elucidate their paleobiology and evolution (*Kruta et al., 2011*; *Hoffmann et al., 2014*; *Lemanis et al., 2015*; *Naglik et al., 2015*; *Tajika et al., 2015a*; *Tajika et al., 2015b*; *Inoue & Kondo, 2016*; *Lemanis et al., 2016*; *Lemanis, Zachow & Hoffmann, 2016*; *Takeda et al., 2016*; *Stilkerich, Smrecak & De Baets, 2017*; *Hoffmann et al., 2018*; *Lemanis & Zlotnikov, 2018*; *Tajika et al., 2018*; *Morón-Alfonso, 2019*). Although tomographic methods allow for detailed descriptions of cephalopod conchs through ontogeny, the number of ontogenetic points needed to accurately describe a conch is unclear. Studying a low number of ontogenetic points may produce an inaccurate growth trajectory, thereby obscuring finer details of ontogenetic change, whereas examination of too many ontogenetic points increases the workload required for data collection and analysis. Insufficient ontogenetic sampling may also bias the results of various studies such as those pertaining to intraspecific variation. To date, researchers have employed different resolutions of ontogenetic sampling when conducting morphometry in cephalopod conchs. For instance, *Hoffmann, Reinhoff & Lemanis (2015)* took measurements of an internal conch of the modern squid *Spirula spirula* (Linnaeus, 1758) every 10°. *Tajika et al. (2018)* examined conchs of the modern nautilid *Nautilus pompilius* Linnaeus, 1758 using 45° intervals. *Morón-Alfonso (2019)* employed 30° intervals to study the Cretaceous ammonite *Maorites*.

*Nautilus* has been studied as a reference to understand the palaeobiology of extinct ectocochleate cephalopods (e.g., *Ward, 1987*; *Saunders & Landman, 2009*). Understanding of their intraspecific variation, ecology, phylogeny, and evolution is expected to improve with the use of tomographic techniques.

In this paper, we present morphological data taken from the intervals of 10°, 30°, 45°, 90°, and 180° to discuss the effect of different sampling densities on the accuracy of projected growth trajectories. Additionally, we briefly assess the requirements for different palaeobiological research questions that can be addressed using these data.

## MATERIALS & METHODS

We examined a single adult specimen of *Nautilus pompilius* (PIMUZ 7825; reposited at the Palaeontological Institute and Museum, University of Zurich). The specimen was collected in Southeast Asia (exact locality unknown). The relatively narrow and rectangular aperture suggests that the specimen is male. The conch diameter is 167 mm,

A computed tomogram of *N. pompilius* was acquired using a Nikon XT H 225 ST industrial CT-scanner at the University of Zurich with the data acquisition parameters of 186 kV and 387 mA. This resulted in volume data sets (TIFF-stack) with the isotropic spatial resolution of 0.0906 mm. Assuming the siphuncle is positioned in the plane of

symmetry, the median section in the image stack was produced to measure the following conch parameters (Fig. 1F): diameter (dm), apertural height (ah), whorl width (ww), and distance between the ventral edge of the siphuncle and the ventral edge of the conch (vd). These parameters are commonly used for cephalopod morphological descriptions (*Korn, 2010*; *Klug et al., 2015*; *Landman et al., 2018*). On the basis of the parameters measured above, we calculated the following ratios: whorl expansion rate WER = $(dm_1/dm_2)^2$, whorl width index WWI = $ww_1/dm_1$, and siphuncle position index SPI = vd/ah. Additionally, the number of septa was counted every 180° (septal spacing index: SSI; Fig. 1E). Measurements of the conch were taken at intervals of 10°, 30°, 45°, 90°, and 180° (Figs. 1A–1E). When measurements are taken every 10°, 110 ontogenetic points are produced, which is the highest sampling density in this study. These linear measurements were taken, evaluated, and visualized using the Image Processing Toolbox of MATLAB 2019a (MathWorks).

## RESULTS

The calculated conch parameters are shown in Figs. 2–5 and Table S1. Due to large differences between maximal and minimal values in early ontogeny and relatively stable values in later ontogeny in WER and WWI, the patterns in late ontogeny are not clearly visible in Figs. 2A, 2C, 2E, 2G, , 2I and 3A, 3C, 3E, , 3G, 3I. Therefore, the trajectories in later ontogeny were magnified (Figs. 2B, 2D, 2F, 2H, 2J and 3B, 3D, 3F, 3H, 3J).

There is a critical point in growth trajectories of conch parameters, at which each value reaches an approximate plateau (at a conch diameter of ∼25 mm; Figs. 2 and 3). In addition, growth changes toward the end of ontogeny are visible in all parameters. These growth changes most likely indicate the "morphogenetic countdown" according to *Seilacher & Gunji (1993)*, although they mentioned that the morphogenetic countdown in *Nautilus* is visible only in colouration. The points at which the growth changes— morphogenetic countdown—start slightly differ between the parameters (Fig. 5). In addition, the colouration extinguishes at a diameter of ∼115 mm, which appears to differ from the conch parameters examined.

WER decreases until a diameter of approximately 25 mm, followed by a relatively stable value at about 2.8 (Fig. 2A). At a diameter of about 100 mm (when measured every 10°; Figs. 2A, 2B), WER shows an increase to reach nearly 3.0. WWI displays a sharp increase until a conch diameter of around 8 mm, followed by a rapid drop until a conch diameter of roughly 25 mm. The value then becomes roughly stable at 0.53 in later ontogeny. The morphogenetic countdown in WWI starts at a diameter of approximately 140 mm (Figs. 3A, 3B). This reveals that the timing at which the morphogenetic countdown starts is not shared between WER and WWI. SPI rapidly increases until a conch diameter of 10 mm and it then sharply decreases until a conch diameter of approximately 20 mm (Fig. 4A). SPI increases again reaching a value of ∼0.47. Subsequently, it remains stable, showing some fluctuating trend until a diameter of 70 mm. Throughout the rest of ontogeny, SPI decreases, most likely indicating the onset of the morphogenetic countdown. A similar ontogenetic pattern occurs in SSI (Fig. 4B): there are phases of sharp increases and decreases

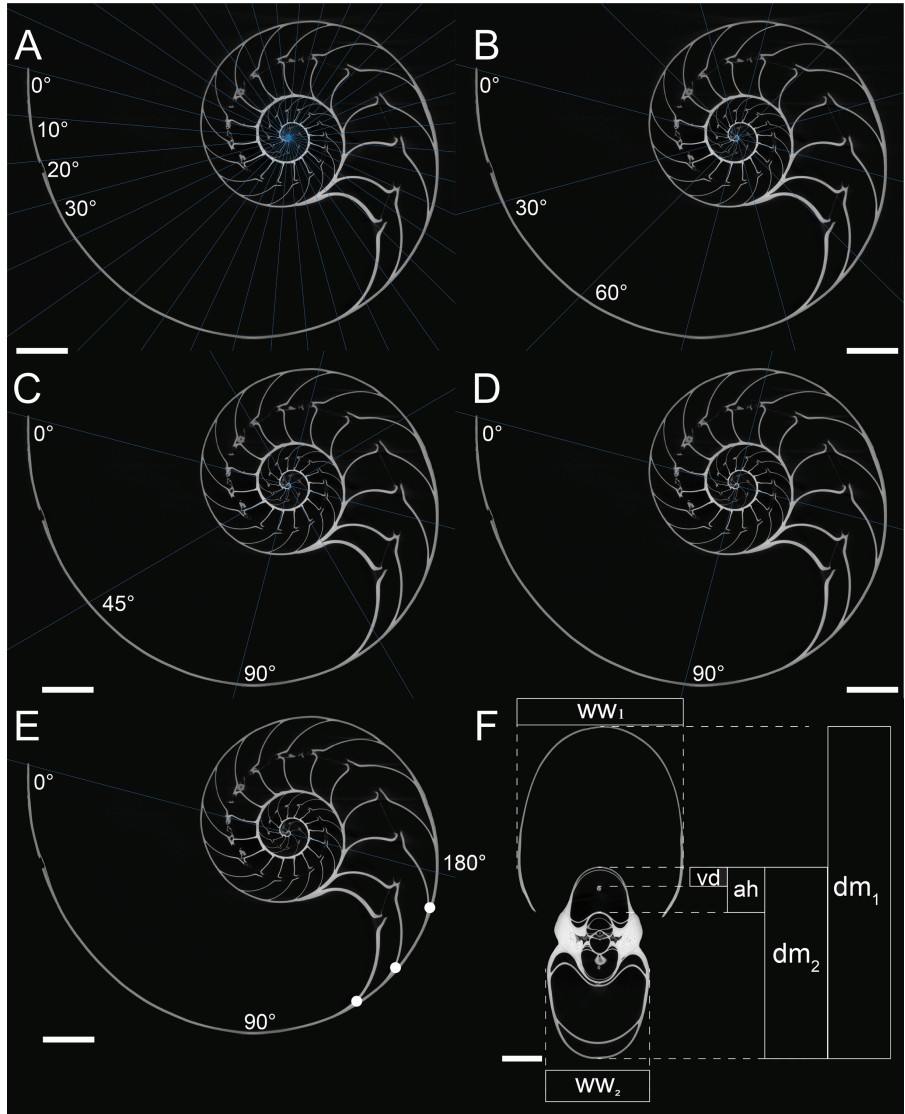

**Figure 1  Measurements of conch parameters on CT-images of the studied *Nautilus pompilius*.** (A) Measurements for every 10°. (B) measurements for every 30°. (C) measurements for every 45°. (D) measurements for every 90°. (E) Measurements for every 180°. Dots represent the number of septa per 180° (septal spacing index). (F) Measured conch parameters (dm, conch diameter; ww, whorl width; ah, apertural height; vd, distance between the ventral edge of the siphuncle and the ventral edge of the conch). Angles represent intervals at which measurements were taken (with 0° as the start point of drawing the lines). Scale bars are 20 mm.

in the earliest ontogeny, followed by a gradual increase until a conch diameter of 30–40 mm. SSI is stable at a value of about eight until it begins to decrease at a diameter of roughly 120 mm (morphogenetic countdown). All ontogenetic trajectories with different intervals are compared in Fig. 5.

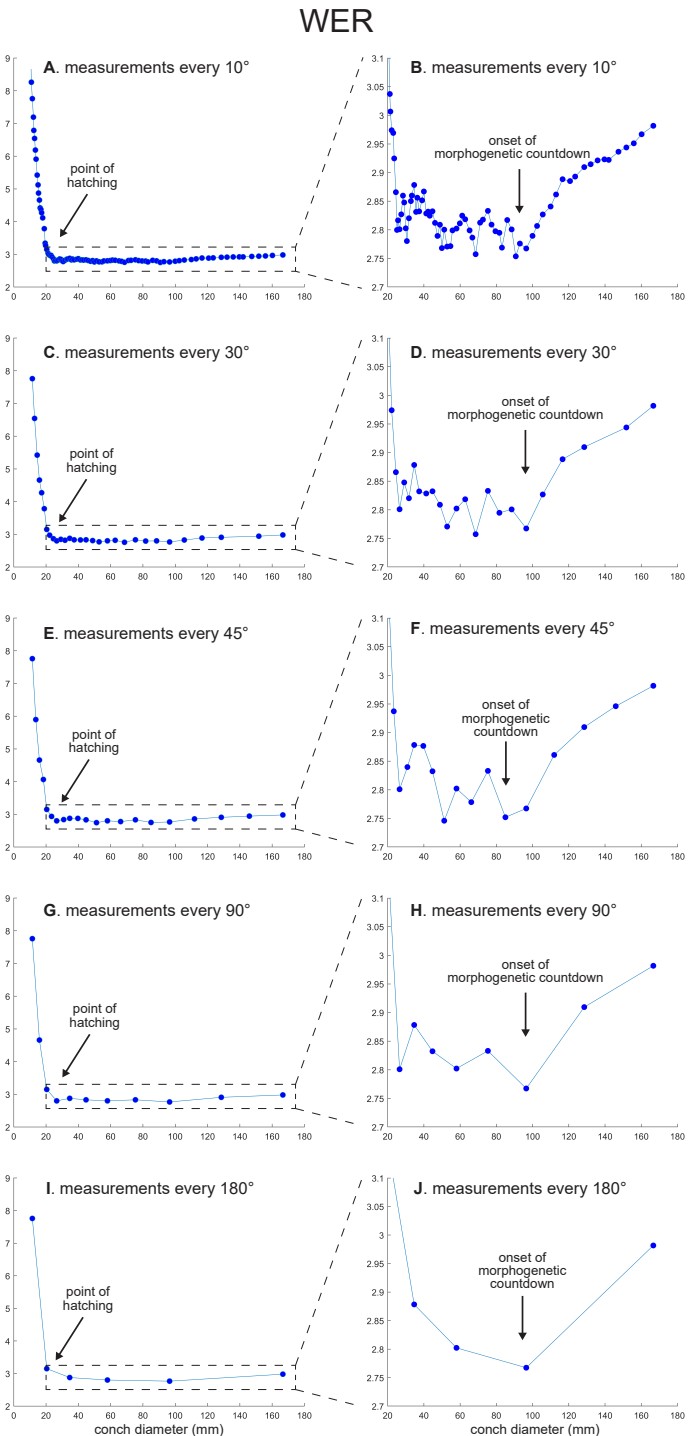

**Figure 2** **Ontogenetic trajectories of the whorl expansion rate (WER).** (A) Trajectory when measuring the conch every 10°. (B) Magnified detail of the graph in A. (C) Trajectory when measuring the conch every 30°. (D) Magnified detail of the graph in C. (E) Trajectory when measuring the conch every 45°. (F) Magnified detail of the graph in E. (G) Trajectory when measuring the conch every 90°. (H) Magnified detail of the graph in G. (I) Trajectory when measuring the conch every 180°. (J) Magnified detail of the graph in I.

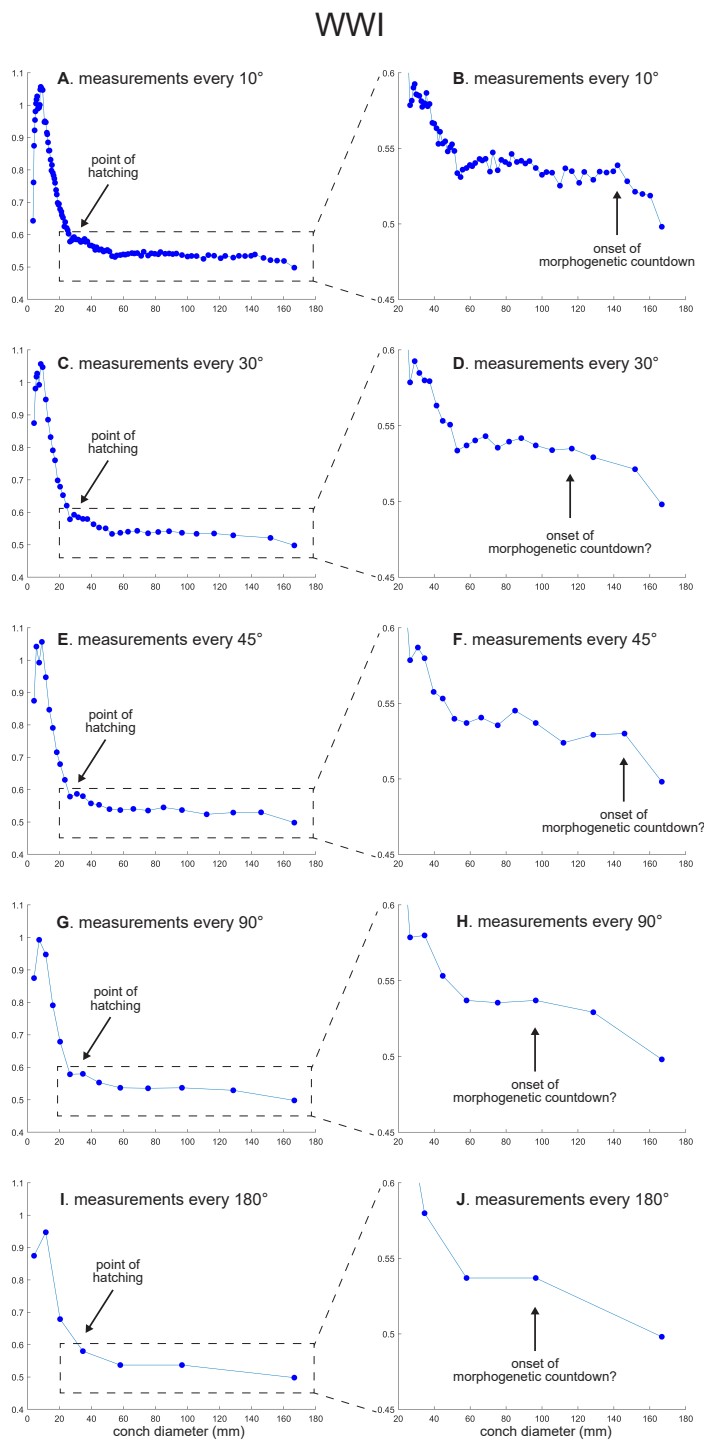

**Figure 3** **Ontogenetic trajectories of the whorl width index (WWI).** (A) Trajectory when measuring the conch every 10°. (B) Magnified detail of the graph in A. (C) Trajectory when measuring the conch every 30°. (D) Magnified detail of the graph in C. (E) Trajectory when measuring the conch every 45°. (F) Magnified detail of the graph in 3E. (G) Trajectory when measuring the conch every 90°. (H) Magnified detail of the graph in G. (I) Trajectory when measuring the conch every 180°. (J) Magnified detail of the graph in I.

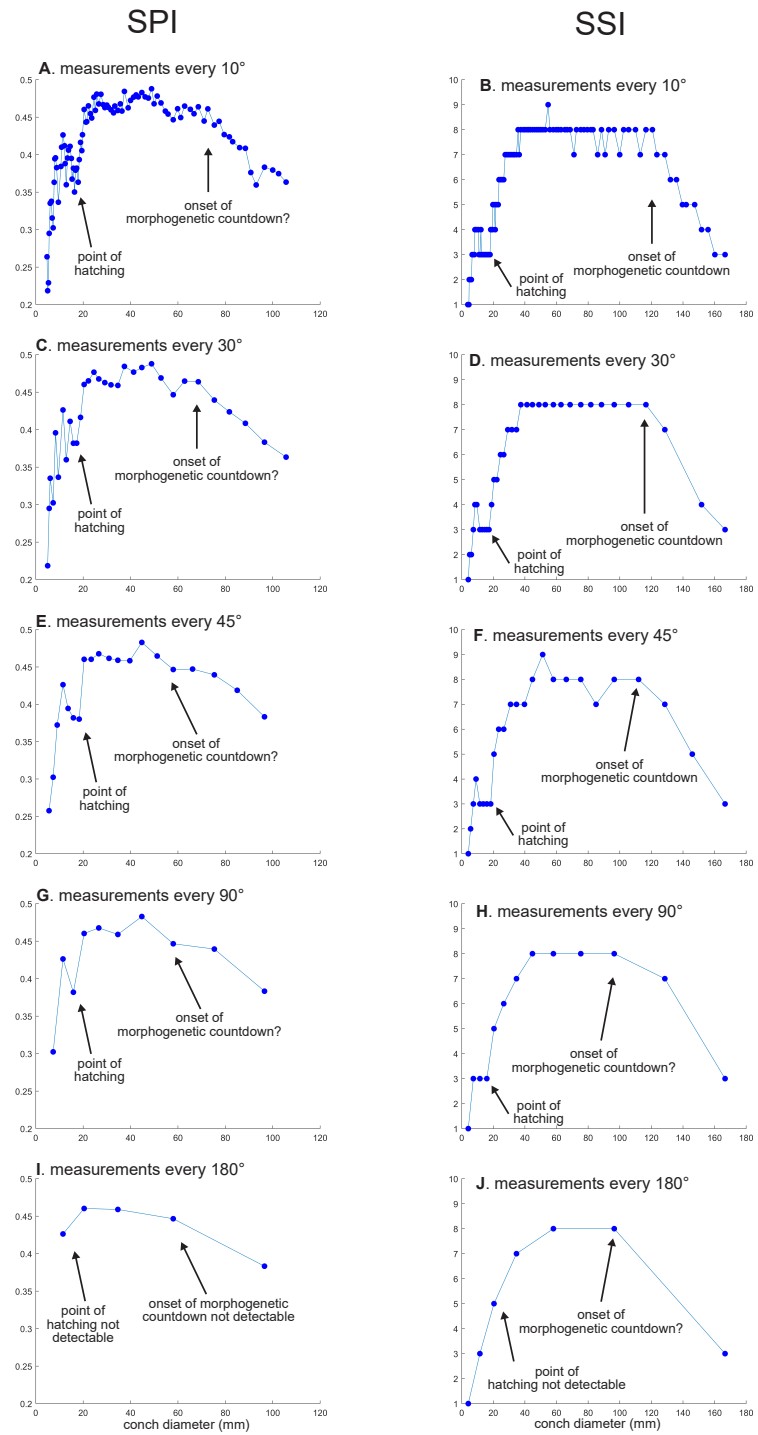

**Figure 4  Ontogenetic trajectories of the siphuncle position index (SPI) and the septal spacing index (SSI).** (A) Trajectory in SPI when measuring the conch every 10°. (B) Trajectory in SSI when measuring the conch every 10°. (C) Trajectory in SPI when measuring the conch every 30°. (D) Trajectory in SSI when measuring the conch every 30°. (E) Trajectory in SSI when measuring the conch every 45°. (F) Trajectory in SSI when measuring the conch every 45° . (G) Trajectory in SPI when measuring the conch every 90°. (H) trajectory in SSI when measuring the conch every 90°. (I) Trajectory in SPI when measuring the conch every 180°. (J) Trajectory in SSI when measuring the conch every 180°.

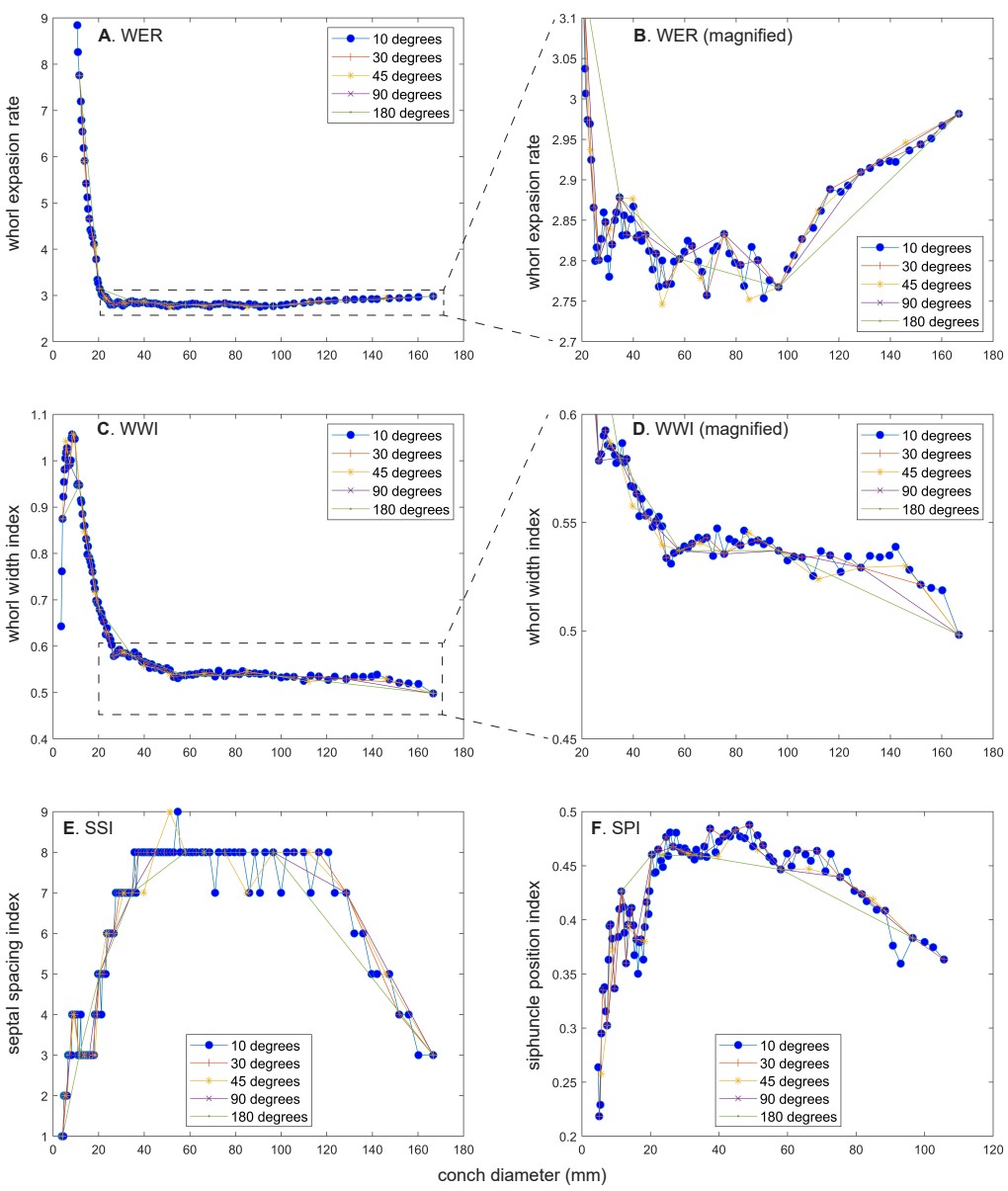

**Figure 5 Comparison of ontogenetic trajectories with different sampling densities.** (A) Whorl expansion rate (WER). (B) Whorl expansion rate (WER) without early ontogeny. (C) Whorl width index (WWI). (D) Whorl width index (WWI) without early ontogeny. (E) Septal spacing index (SSI). (F) Siphuncle position index (SPI).

## DISCUSSION

### Accuracy of conch trajectories with different sampling densities

Two critical points at which growth changes occur are shared between the ontogenetic trajectories of all examined parameters. The first conspicuous growth change occurs at a diameter of ~20–25 mm. This growth change most likely indicates the point of hatching in *Nautilus* (*Arnold, Landman & Mutvei, 1989*). The second distinct growth change occurs
shortly before the attainment of maturity (morphogenetic countdown), although the timing of the onset of this morphogenetic countdown appears to vary among the conch parameters. In the following section, we discuss the effects of sampling density on the accuracy of growth trajectories in each conch parameter.

*WER*: Comparing the ontogenetic trajectories of WER at different sampling densities reveals roughly similar patterns (Fig. 2). There is a subtle difference in the timing at which the abovementioned growth changes occur between the trajectories with different sampling densities (Fig. 5); finer details tend to be masked in the trajectories with lower sampling densities. For example, the first growth change (hatching event) appears to happen at about a conch diameter of 25.2 mm (measurements every 10°; Fig. 2A), the point at which it occurs is shifted to a conch diameter of about 20 mm (measurements every 180°; Fig. 2I). Furthermore, roughly four cycles of increases and decreases between the diameters of 20 and 100 mm are observed in Figs. 2B, 2D and 2F (measurements every 10°, 30°, and 45°). These cycles are barely visible in Fig. 2H (measurements every 90°) and invisible in Fig. 2J (measurements every 180°). Lastly, there is a slight growth change during the morphologic countdown at a conch diameter of approximately 120 mm (Figs. 2B, 2D, 2F). This growth change is not documented in Fig. 2J (measurements every 180°) and the timing at which it occurs is shifted to a larger conch diameter in Fig. 2H (measurements every 90°).

*WWI*: As in WER, the overall pattern is similar in the trajectories with different sampling densities (Figs. 3 and 5). A large error in the timing of hatching occurs when measurements are taken at the intervals of 180° where the point at which the growth change happened is shifted from about 26 mm (measurements every 10°) to 34 mm (measurements every 180°). In addition to the growth change at the point of hatching, there is another growth change at a conch diameter of 6–8 mm in WWI (Fig. 3A), which is first documented here. This growth change may be linked to the profound growth change from a flat to a curved shape, which indicates the end of the "metanepionic stage" where a slight construction may appear (*Arnold, Landman & Mutvei, 1987*). Although this growth change is visible in all trajectories, it is less clear when using a lower sampling density (Figs. 3G, 3I). The fluctuation in middle ontogeny between the conch diameters of 40 and 140 mm (Fig. 3B) is also masked in Figs. 3H and 3J (measurements every 90° and 180°). Figure 3B shows that the morphogenetic countdown starts at a diameter of approximately 140 mm but the timing appears to differ in the other trajectories (Figs. 3D, 3F, 3H, 3J and 5).

*SPI*: Before hatching, there is a point at which a growth change occurs in SPI. This growth change is well documented except when using 180° intervals (Fig. 4I). In addition, Fig. 4I does not clearly illustrate the overall pattern because the rapid increase in the earliest ontogeny is missing. The overall trend appears to be shared except when using 180° intervals (Fig. 4I). The onset of the morphogenetic countdown is rather difficult to detect in SPI; this is because the onset of decrease related to morphogenetic countdown is barely distinguishable from some general decreasing trend that fluctuates in middle ontogeny (Fig. 4A). Nevertheless, a sharp drop, which occurred during the terminal countdown at a diameter of 90 mm in Fig. 4A, is not documented in the other trajectories (Figs. 4C, 4E, 4G, 4I and 5).

*SSI*: As in WWI and SPI, a growth change occurs before hatching in SSI. This corroborates that a pre-hatching growth change is common in most conch parameters in *N. pompilius.* The overall ontogenetic pattern appears to be similar among all trajectories (Figs. 4B, 4D, 4F, 4H, 4J and 5). There tend to be relatively large errors in the timing of growth changes when the sampling density is lower. For instance, the first growth change (hatching event) is visible at a conch diameter of approximately 20 mm (Figs. 4B, 4D, 4F, 4H). But this growth change is not detectable in Fig. 2J (measurements every 180°). In Figs. 4B and 4D, (measurements every 10° and 30°), SSI reaches a plateau at a conch diameter of ∼30–40 mm, whereas the growth change occurs at a seemingly slightly larger diameter in Figs. 4H and 4J (measurements every 90° and 180°). Additionally, the onset of the morphogenetic countdown happens at a smaller diameter in the trajectories with lower sampling densities (Figs. 4H, 4J). Our data on WER, WWI, SPI, and SSI suggest that relatively large errors in growth trajectories occur more often when measurements are taken every 90° and 180°.

## Research question vs. choice of sampling resolution

While our results suggest that measurements taken solely at demi-whorls are sufficient in revealing overall ontogenetic trends, finer details may be lost by poor resolution sampling. Namely, relatively large errors occur at the sampling densities of 90° and 180° (i.e., the timing of hatching is shifted and short-time fluctuation is lost; e.g., Figs. 2H, 3I, 3J). These fine details may concern the exact placing of events such as hatching, mature growth, or growth disturbances caused by injuries, adverse conditions such as low oxygen conditions, poor food availability and toxic chemical composition of the seawater, epizoan overgrowth etc. (e.g., *De Baets, Keupp & Klug, 2015*; *Tajika et al., 2015b*). Accordingly, when addressing research questions regarding the placement of growth changes and/or reconstructing the effects of disturbances, a higher resolution is necessary (optimally, at angular increments of ≤30°).

## Measurements in other taxa

Naturally, the angular spacing of changes in coiling and hence growth parameters is dependent on the coiling itself (i.e., on the mode of coiling). For example, changes in septal spacing can be determined proportionally more accurately at the same angular increments in an ammonoid that formed ten whorls than in a nautiloid that formed only two whorls during its entire development. This implies that sampling must be adapted to the mode of coiling. Additionally, growth is not uniformly homogeneous or heterogeneous in all coiled cephalopod conchs. *Tajika et al. (2020)* revealed that planispirally coiled ammonoids are generally more responsive to environmental perturbations than nautilids in their chamber volume development. This suggests that short-term fluctuations in growth parameters vary profoundly in intensity between the two groups. This is of relevance because these rapid fluctuations are lost when large, angular increments of measurements are employed. Consequently, we recommend choosing smaller angular increments (<30°) when examining ammonoid ontogeny. It appears that nautilid conchs grow more harmonically and exhibit less intense, short-term fluctuations, thus allowing

for larger angular increments. In any case, it is probably safe to examine one specimen of the taxon of interest to test possible errors resulting from different sampling resolutions, not resolution before working with multiple specimens.

## CONCLUSIONS

We investigated the effects of different sampling densities on the resultant growth trajectory in *Nautilus pompilius*. By examining the conch using a high sampling density, we discovered a previously undocumented growth change that occurs before hatching. Additionally, we found that the points at which the morphogenetic countdown starts differ between various conch parameters.

When studying a fewer number of ontogenetic points, this growth change is much less conspicuous. Furthermore, the timings at which growth changes occur vary among the trajectories when different sampling densities are applied; accuracy is diminished at larger angular increments. Fluctuations of various conch parameters in middle ontogeny are often masked in ontogenetic trajectories projected using a lower sampling density. We found that these errors are especially large in certain parameters when measurements are taken every 90° and 180°. In contrast, trajectories are reasonably similar when studying a conch every 10°, 30°, and 45°. The timing at which growth changes and fluctuations occur differs among various cephalopod taxa and, therefore, testing the effect of ontogenetic sampling density is necessary. Accordingly, we recommend choosing low angular increments of 30° or less when examining the effects of growth disturbances and for studies regarding the exact placement of ontogenetic changes (hatching or the onset of the morphogenetic countdown). However, the stronger short-term fluctuations of growth parameters in ammonoids require a denser sampling of measurements through ontogeny than most nautilids.

## ACKNOWLEDGEMENTS

We thank Shannon K. Brophy (Brooklyn College) for proofreading the manuscript. The reviews by Ryoji Wani (Yokohama National University), Romain Jattiot (Universität Bremen), and Jérémie Bardin (Sorbonne Université) are greatly appreciated.

### Funding

This work was supported by the Japan Society for the Promotion of Science and the Swiss National Science Foundation (project nr. 200021_169627). There was no additional external funding received for this study. The funders had no role in study design, data collection and analysis, decision to publish, or preparation of the manuscript.

### Grant Disclosures

The following grant information was disclosed by the authors:
Japan Society for the Promotion of Science.
Swiss National Science Foundation: proj 200021_169627.

## Competing Interests

The authors declare there are no competing interests.

## Author Contributions

- Amane Tajika conceived and designed the experiments, performed the experiments, analyzed the data, prepared figures and/or tables, authored or reviewed drafts of the paper, and approved the final draft.
- Christian Klug analyzed the data, authored or reviewed drafts of the paper, and approved the final draft.

## Data Availability

The data is available at MorphoSource: M54790-107483.

## Supplemental Information

Supplemental information for this article can be found online at http://dx.doi.org/10.7717/peerj.8849#supplemental-information.

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
