# Peer review of "How many ontogenetic points are needed to accurately describe the ontogeny of a cephalopod conch? A case study of the modern nautilid Nautilus pompilius"

_PeerJ, doi:10.7717/peerj.8849_

## Round 0.1 · original submission · Minor Revisions

Dear Author,

Please find attached the reviews of your ms. The comments of reviewers 1,2 seem to me to be the most pertinent. Reviewer 3 raises some important points but I suspect you could counter by saying that they (esp. suggestion 2) are beyond the scope of the study. However, I think it could be useful at least to discuss in more detail the issues surrounding use of multiple specimens and how to deal with error. Discussions of these concepts would extend the applicability of your results, for example to ecologists wanting to assess growth defects or, at least, point to further research possibilities.

Overall, I concur with Reviewers 1,2 in suggesting Minor Revision. I look forward to receiving the revised version, and can provide some (minor) suggestions on improving readability at that point.

·

Basic reporting

I feel that this manuscript is well designed to PeerJ. English is very clear (at least I feel so), because this manuscript is easily understandable even for non-native speaker (=me). The manuscript has sufficient introduction and background to demonstrate. Figures are beautiful, which is relevant to the content of the manuscript. The raw data is shown in supplementary file. Therefore, in this point of view, the manuscript totally follows the standards of PeerJ.

Experimental design

I feel that the experimental design of this manuscript is well organized. The investigation of the manuscript was conducted rigorously with a high technical standard. Therefore, in this point of view, the manuscript totally follows the standards of PeerJ.

Validity of the findings

I found one point that should be revised, to increase the robustness.
(1) Is the examined specimen male or female?
I think that the information of male or female is important, because this study examines the whorl width of the shell. Nautilus shells show sexual dimorphism, which is well recognized in the difference of whorl width. Therefore, I’d like to request to add the information of sex of the examined specimen, if possible. As far as I see the cross section of the specimen in Fig. 1F, the specimen seems to be male.

(2) Information of shell diameter of the examined specimen
It is better to mention the shell diameter of the examined specimen in Material section (lines 73–75). We can recognize from the figures that the shell diameter of the examined specimen is around 170 mm. However, I think it is much fair to mention in the main text.

(3) The definition of Septal spacing index (SSI)
As far as I recognize from Fig. 1E, septal spacing index (SSI) at the last preserved aperture is eight. If so, it means this index represents the septal number of the “previous” half-whorl. I feel this is strange. I guess that this index should be counted and calculated in the “last” preserved half-whorl at each measured aperture point.

(4) Different shell diameter of the timing of the onset of morphogenetic countdown
This study reveals the timings of the onset of morphogenetic countdown are different among the different parameters of shells (WER, WWI, SPI, SSI). As far as my recognition, such difference of the timings of the onset of morphogenetic countdown has never been documented and demonstrated. Therefore, I’d like to request to the sentence of lines 123–124 should be more emphasized or the authors prepare another manuscript to fully demonstrate such difference. Relating this, I’d like to know whether the timing of morphogenetic countdown of shell coloration (the presence or absence of brown stripe coloration) is concordant to the timings in this study. Seilacher and Gunji (1993) mentioned the morphogenetic countdown in modern Nautilus only in shell coloration. I expect that the authors can discuss more fully with 3D data the differences of timings of the onset of morphogenetic countdown among plural different parameters of shells, which might be better to prepare another manuscript.

(5) Shell diameter at which morphogenetic countdown starts
The authors mentioned that the shell diameters at which morphogenetic countdown starts in WER are different, depending on measurement intervals (lines 136–138). I agree that “the timing at which it occurs is shifted”, however I’m not sure that the shift is always toward “larger conch diameter”. I think that larger or smaller depends on where the measurement point with 90° interval is situated. In the case of this manuscript (Fig. 2H; the shell diameter of measured point is around 100 mm), the shift is toward “larger” conch diameter. However, if the measurement point with 90° interval is situated a little bit smaller shell diameter (e.g., about 90 mm in shell diameter), the shift would be toward “smaller”.
This is same for SSI case (line 166). The authors mentioned “the growth change occurs at a seemingly slightly larger diameter in Fig. 4H and 4J”, however, I feel these also depend on how the measurement points with 90° or 180° intervals are settled.
I’d like to request to add more explanation how the measurement points are settled. Then, if the authors want to mention that the shift is toward “larger” or “smaller”, the authors should confirm in advance that the shifts are always (not depending on how the measurement points are settled) “larger” or “smaller”.

(6) shell shape change before hatching
The authors found that the shell shape changes before hatching. This is very interesting and new finding. The authors thought that this shell shape change “might be linked to the profound growth change from the conical, Patella-shaped initial conch” (lines 143–144). This is very interesting suggestion. I guess that there are some previous studies about shell formation within egg capsule in modern Nautilus (possibly in “Nautilus” book in 1987 from the Plenum Press).
I’d like to know whether the shell diameters at which the previously unknown shell change occur are concordant to the shell size “growth change from the conical, Patella-shaped initial conch” in the literatures.

(7) The fluctuation in middle ontogeny
The authors also found that the fluctuation in middle ontogeny in WER and WWI. I’d like request to add more suggestions what kind(s) of ontogenetic event(s) are related to such fluctuation of shell shape.

(8) A sharp drop of SPI at a diameter of 90 mm in Fig. 4A
The authors supposed that the onset of morphogenetic countdown in SPI is situated around 70 mm in shell diameter (Fig. 4). However, is a sharp drop of SPI at a diameter of 90 mm in Fig. 4A the onset of morphogenetic countdown in SPI?

(9) Suggestion for ammonoids
I feel that the discussion about “Measurements in other taxa” is not enough. The authors mentioned “For ammonoids, a much denser sampling may be required” (lines 36–37, 191–193, 210–211). However, I cannot fully understand why a much denser sampling is better in ammonoids. The authors cited their submitted manuscript (Tajika et al., submitted) for the reasons. However, we cannot understand the details of Tajika et al. (submitted). How many ammonoids were analyzed? What geological ages, Paleozoic, Mesozoic, or both? How is shape of the examined ammonoids, normally coiled ammonoids or heteromorphs? If normally coiled ammonoids, how is shape, similar to those in modern Nautilus? How is the completeness of the examined ammonoid sutures? I think, without such information, we cannot understand why the authors conclude that a much denser sampling is better in ammonoids. Therefore, I’d like to request that enough information to recognize why a much denser sampling is better in ammonoids should be mentioned in the revised manuscript.

(10) Several minor suggestions
Line 130: “abovementioned” should be “above mentioned”.
Line 176: I think “adverse conditions” is ambiguous. Please mention it more specifically.
Line 179: Please add “°” after 30.
Line 242: Please add journal information.
Line 290: “Ontogenetic trajectories of the septal position index (SPI)” should be “Ontogenetic trajectories of the siphuncle position index (SPI)”
Fig. 4B: The position of “onset of morphogenetic countdown” seems to be a little bit larger from the peak at 120 mm.
Fig. 4B, D, F, H: Please add arrows showing the position of “point of hatching”.

Additional comments

This manuscript is of great interest for many paleontologists, especially cephalopod workers, because it provides an interesting information how many ontogenetic points are needed to precisely describe the ontogeny of modern Nautilus conch. It contains great possibility to apply various cephalopod shells (nautiloids, ammonoids). I made some comments that should be revised, to increase the robustness. However, most of them are trivial issues, so that I think the authors can revise the manuscript easily. Therefore, I would like to strongly recommend accepting this interesting manuscript for the publication in PeerJ, with minor revision.

·

Basic reporting

The paper is well written with clear and very professional English. The figures are good and straightforward. The raw data are shared. The results are very relevant towards providing answers to the work questions/hypotheses.

Regarding the field background, the authors state that the number of studies using computed tomography has increased dramatically over past decades. Then they give a list of 8 works on cephalopods, particularly (L. 49). Of those, I am just surprised that there seem to be very few that worked on tomographic reconstructions of the living Nautilus conch. It is just a personal comment, I’m very confident that they have all the literature in mind. I was just expecting more, but this makes their work even more interesting!
Nevertheless, I think the authors could add some more references that performed tomography to elucidate the morphology, ontogeny, paleobiology and evolution of cephalopods. For instance, I think that L. 49-51 they could cite Inoue & Kondo 2016 who worked on ammonite suture line growth using tomography; and I’m guessing there should be more of this kind to cite.

Inoue, S., & Kondo, S. (2016). Suture pattern formation in ammonites and the unknown rear mantle structure. Scientific reports, 6, 33689.

Overall, since there are less than 20 references in total (15 if excluding the ones from the first author), I recommend that the authors try to find a bit more related works, if possible and if within the frame of the work, naturally (to be cited in the introduction and in the two last parts of the discussion).

Experimental design

The research questions are remarkably well-defined. It is explicitly stated how this work is going to fill an identified knowledge gap.

Two small questions remain:

Why did you choose to measure every 10, 30, 45, 90 and 180 degrees? From 45, you chose to double the degrees, why not do the same before so that the difference between each interval is similar? (i.e., about 20° instead of 30°?).

Did you test the degree of measurement errors by taking the same measurements 3 times for the same specimen as you did in your 2018 paper?

Tajika, A., Morimoto, N., Wani, R., & Klug, C. (2018). Intraspecific variation in cephalopod conchs changes during ontogeny: perspectives from three-dimensional morphometry of Nautilus pompilius. Paleobiology, 44(1), 118-130.

Validity of the findings

The findings are very robust.

I was just a bit puzzled about one thing when you write L. 31 (and a few times elsewhere) that “a previously undocumented growth change before hatching was detected when measurements were taken every 10°, 30° and 45°”: Why was this change not documented in your previous work (Tajika et al. 2018 in Paleobiology)? If I am not mistaken, you have also reconstructed ontogenetic trajectories in this work, with measurements from 45° intervals as you stated in the methodology of this paper.

Regarding the discussion on the effects of sampling density on the accuracy of growth trajectories (i.e., by graphically comparing the ontogenetic trajectories at different sampling densities), don’t you think it would possible for you to precisely determine the differences in the shape of the graphs depending on the intervals taken? Perhaps there is a statistical way to quantify the differences between two graphs. Something that would tell you, for instance, that WER graph 10° shares 90% of similarities with SSI graph 30°. What I can think about is for example to calculate the surface of each curve and to compare them or to calculate an overlap percentage.

Alternatively, wouldn’t it be possible that even with a figure superimposing the 5 graphs (each with a different color and different degree of transparency), the reader could see where the main differences lie? It might be worth trying.

Additional comments

Overall, the manuscript is well structured: the working hypotheses are particularly crystal-clear; so is the methodology. However, it just seems to me that the first 5 paragraphs of the discussion might fit better as results, if it is doable. I suggest this based on the near absence of references and real discussion in those paragraphs and also based on the way the graphs are routinely compared in them.

Furthermore, one general issue to me is that I was expecting a slightly more developed discussion, given the interesting methodology and results. At least, I recommend that the authors clarify some parts of the discussion. For instance, they write L. 174 that finer details may be lost by poor resolution sampling, such as the hatching event. This is surprising because in all cases (10, 30, 45, 90 and 180°), they could document the hatching at nearly the same moment (about 20 mm; Fig. 2). I acknowledge that with 180° interval it is not the exact same moment, so it might be good to specify that “poor resolution” means 180°. I think that there are in the discussion too many unclear or uninformative terms such as "more or less" (l. 173), “smaller”, “larger”, “poor”. I strongly recommend that the authors more strongly link the discussion to their results.
This might actually be solved with a restructuring of the paragraphs included in the discussion, as I commented already above, perhaps?


In conclusion, I have great confidence that this paper overall meets the standards for publication in PeerJ; and I therefore believe that this paper will be acceptable after minor revision.

Here are my minor suggestions-only to the authors:

→ Title: I suggest “How many ontogenetic points are needed to adequately/suitably/satisfactorily/accurately describe the ontogeny of a cephalopod conch?”, the way you actually wrote it L. 23 and L. 53.
→ Title: “living” instead of “modern” might be even more straightforward
→ L. 18: “ morphological analysis” in plural?
→ L. 24: replace “the effect” with “the impact”?
→ L.26: replace “trajectories of a conch” with “trajectories of one/a single conch”, i.e., the way you wrote it L. 73 “a single adult specimen”
→ L. 36: Could you already in the abstract briefly give some reasons why “for ammonoids, a much denser sampling may be required”?
→ L. 44: “yet” or “although” might be here more appropriate than “whereas”
→ L. 46: “ideal model group”. I think one of these two terms is enough: “cephalopods are an ideal group” or “cephalopods are an ideal model”. I would also right after this sentence add what you wrote in the abstract, i.e., write that this is because the entire ontogeny is visible within a conch.
→ L. 52: “throughout ontogeny”?
→ L. 67: regarding the intervals, see my comment in the “experimental design” section
→L. 68: replace “the effect” with “the impact”?
→ L. 73: I suggest you write “Linnaeus, 1758” with the very first mention of Nautilus pompilius, i.e., L. 61.
→ L. 75: “..collected in Southeast Asia (exact locality unknown)”
→ L. 83: “(e.g., Korn 2010…) since there many other works that could be cited here.
→ L. 85: “WWI = ww/dm1”, isn't it?
→ L. 86: Shouldn’t it be “ah” instead of “wh” in “vd/wh”. I don’t see any “wh” in Figure 1F. Furthermore, you write “additionally, the number of septa was counted every 180° and here I get confused since in Figure 4, there is a SSI graph for each case (10°, 30°, 45°, 90° and 180°).
→ L. 94: I am not sure the word “gaps” fits the best here. I only suggest “differences” or “disparity”.
→ L. 98: “There is a critical point….plateau” You might want to call Figure 2 and 3 here.
→ L. 102: “WWR decreases until a diameter of approximately 25 mm, followed by a relatively stable value at about 2.8. Is this regardless of the intervals taken for measurements? In case you are referring to a specific figure here, you should call it (as done in the next sentence). Is this value of 2.8 of importance? If yes, shouldn’t you add a horizontal line on the graphs to represent this mean value?
→ L. 103: “Before the attainment of maturity at a diameter of about 100mm…WER show an increase to reach nearly 3.0” As it is, the sentence is misleading. The reader could understand that the attainment of maturity occurs at a diameter of about 100 mm, which is not the case if I am not mistaken (maturity is reached at a diameter of about 180 mm). I would remove “Before the attainment of maturity” as it is not very informative or write: “At a diameter of about 100mm, WER shows an increase to reach nearly 3.0 at maturity”.
→ L. 104-107: “WWI displays…..becomes roughly stable at 0.53 in later ontogeny” Again, is this regardless of the invervals taken for measurements? Is this value of 0.53 of importance? If yes, shouldn’t you add a horizontal line on the graphs to represent this mean value?
→ L. 107: “The morphogenetic countdown in WWI starts at a diameter of approximately 140 mm”. I suggest you add “140 mm, instead of 100 mm for WER” so that the reader is reminded of the difference in the onset of the morphogenetic countdown depending on the parameters.
→ L. 108: “SPI rapidly increases…” Actually for WWI, WER, SPI and SPI, shouldn’t there be a “the” before? “The SPI” as for “the Septal Spacing Index”
→ L. 110: “showing some fluctuating trend” This is not very informative. Furthermore, couldn’t it be some noise due to measurement errors?
→ FIG. 4A: I am not sure to understand why you think the onset of the morphogenetic countdown is at a diameter of about 75 mm. I don’t see anything peculiar there. In the text, you do not say anything about this value of 75 mm. Since you write that “Throughout the rest of ontogeny, SPI decreases, most likely indicating the onset of the morphogenetic countdown”, you should put it at a diameter of about 50 mm to be consistent.
→ L. 123: “ The second distinct growth change occurs before the attainment of maturity” The word “before” is not very informative. Would it be wrong to write “shortly before”? And then write “However, the timing of the onset of this morphogenetic countdown appears to vary among the conch parameters”.
→ L. 135: “there is a slight growth change… of approximately 120 mm” I suppose this is visible in Figs. 2B, D and F.
→ L. 142: “which is first documented here” See my comment in “Validity of the findings” section.
→ L. 172: I don’t quite understand the title of this section.
→ L. 173: “are more or less sufficient”. I think you shouldn’t use “more or less” here. The aim of your study was precisely to assess when measurement intervals were or were not sufficient. From your results, I think you can assert that measurements taken at demi-whorls are sufficient in revealing overall ontogenetic trends.
→ L. 177: “This follows that when addressing…” Would it be possible to rephrase the beginning of this sentence? It is not clear to me what you mean.
→L. 179: “is desirable” Given what you just wrote in the previous sentence, I think you should replace “desirable” with “mandatory”. Maybe also precise that what you mean with “higher resolution” is a minimum of 45°.. and optimally <30°.
→ L. 188: “…short term fluctuations in growth parameters vary profoundly in intensity between the two groups”. Fluctuations that vary is an unnecessary repetition:. I suggest: “..there are short-term but profound fluctuations in growth parameters between the two groups”.
→ L. 190: it’s better that you cite again “Tajika et al. submitted” at the end of this sentence.
→ L. 191: “smaller” Please be more specific, smaller than what? 45°? 30°?
→ L. 193: “thus allowing for larger angular increments” Again you could be more specific: give here a optimal value, which you actually give in the abstract (45°)
→ L. 198: “when working with a fewer number of ontogenetic points” perhaps?
→ L. 199: replace “the growth change” with “this growth change”
→L. 201 “accuracy is diminished at larger angular increments”. Isn't it is too obvious to be stated? (at least the way it is written - the following sentences are much more informative).
→ L. 207-211: I recommend that somehow you write the information contained in these two sentences as precisely in the discussion. It seems to me that it is more confusing as it is written for the time being in the discussion.

·

Basic reporting

The manuscript adheres to all PeerJ policies as far as I can judge. The ms is well-written and well-organized. I would not judge the English as I am not a native English speaker. Figures are appropriate and helpful. I think few references could be added in the introduction about measuring the ontogeny of cephalopods and the relationship with absolute age such as Dommergues (1988, Lethaia, 21).
Two minor comments:
- “10 mmm” l. 108 should be “10 mm”
- “dimeter” l. 158 should be “diameter”.

Experimental design

The question asked by the authors is in the title: “how many points are needed to describe the ontogeny of a cephalopod conch”. The authors measured one conch of a modern nautilid with several resolutions (by varying the angle between measurements), described the events of the ontogenetic trajectories and for which resolutions they could be observed or not. The final advice is to use 30° or 45°, being the good balance between resolution and data collection time accompanied with a quick discussion on other taxa and problematics.
In my opinion, this work has two major weaknesses. The first one concerns the way the accuracy of the representation of the ontogenetic trajectory is evaluated. As the authors used only one specimen, I would expect the errors brought by the different resolutions to be measured while, throughout the manuscript, only qualitative descriptions of errors are provided. I know that most authors do not take into account ontogeny as precisely as it should be and only describe it qualitatively. However, a paper dedicated to measurement errors of the ontogeny should treat it as precisely as possible. For example, there is, for this specimen, an exact location in ontogeny of each growth change. How far are we from this exact location by using a given angle? Does the error change depending on the phase (location of the first angle)? Should we model ontogenetic stages to get theoretical locations of these growth changes? Is the acceptable degree of error related to the structures, events or stages under study? I think that the quantification of the errors is unavoidable.
Secondly, the procedure undertaken by the authors is not well-suited to answer the problematic that is way wider. As it is, the paper answers that a given resolution is enough to capture some growth changes of the specimen PIMUZ 7825. However, we only have a limited insight of how general this conclusion can be? The authors are clearly aware of this limitation as developed in the sections “Dependence on the research question” and “Measurements in other taxa” but these two sections need to be extended as they are, in my opinion, the main core of the argumentation to treat the problematic. In which way some other specimens, some other species with other growth changes may be measured with a given resolution? Indeed, it is linked to the variability of what you measure and for what goal. For example, consider the angle at which the trajectory pass from a given growth stage to the following one in the ontogeny. I guess that the resolution to be used will be very different if you just want to detect that this change exists in a given specimen (this paper) or if you want to use the variability of this location in a phylogenetic analysis or to recognize dimorphs or anything else. I really expected to find in the paper a attempt of a procedure suitable to a wider variety of situations with quantification of variability. Something like: 1) identify the hypothesis to test, 2) evaluate the variability of the event(s) or stage(s) used in the hypothesis testing, 3) based on the first two points, deduce a reasonable resolution to use and ensure the test of the hypothesis will be robust enough.

Validity of the findings

Several conclusions are not based on the data and analyses from the manuscript as, for example, the fact that fine details may be lost by poor resolutions. It is likely but not tested here. On the opposite, much more can be extracted from the data herein collected, especially some statistics on errors.

Additional comments

I think that the problematic is extremely interesting and deserve some papers. For now, the ms does not support its claims enough. The analyses are not suited to answer the problematic. I hope my comments will not be too discouraging as I really think a proper treatment of this question may be of great interest for the community. Quantifying the errors should not be that time-consuming and I think it would give greater consistency to the paper.

---

## Round 0.2 · accepted · Accept

Dear Author, thank you for addressing thoroughly the comments and suggestions of the reviewers. I have added my own copyedit suggestions on your track change revised version and append a PDF version of that doc to this decision. PeerJ staff will send you the Word doc. You can incorporate these suggestions, where appropriate, while in production. I look forward to seeing the paper published, best Chris